**Data Availability Statement:** All relevant data are within the manuscript and its Supporting Information files.

# Detection of cross-reactive immunoglobulin A against the severe acute respiratory syndrome-coronavirus-2 spike 1 subunit in saliva

**Keiichi Tsukinoki[1]\*, Tatsuo Yamamoto[2], Keisuke Handa[3], Mariko Iwamiya[4], Juri Saruta[1], Satoshi Ino[5], Takashi Sakurai[6]**

1 Department of Environmental Pathology, Graduate School of Dentistry, Kanagawa Dental University, Yokosuka, Kanagawa, Japan, 2 Department of Dental Sociology, Graduate School of Dentistry, Kanagawa Dental University, Yokosuka, Kanagawa, Japan, 3 Department of Oral Biochemistry, Graduate School of Dentistry, Kanagawa Dental University, Yokosuka, Kanagawa, Japan, 4 Department of Clinical Laboratory, Kanagawa Dental University Hospital, Yokosuka, Kanagawa, Japan, 5 Department of Minimal Intervention Prosthodontics, Graduate School of Dentistry, Kanagawa Dental University, Yokohama, Kanagawa, Japan, 6 Department of Maxillofacial Radiology, Graduate School of Dentistry, Kanagawa Dental University, Yokosuka, Kanagawa, Japan

\* tsukinoki@kdu.ac.jp

## Abstract

Abundant secretory immunoglobulin A (SIgA) in the mucus, breast milk, and saliva provides immunity against infection of mucosal surfaces. Pre-pandemic breast milk samples containing SIgA have been reported to cross-react with SARS-CoV-2; however, it remains unknown whether SIgA showing the cross-reaction with SARS-CoV-2 exists in saliva. We aimed to clarify whether SIgA in saliva cross-reacts with SARS-CoV-2 spike 1 subunit in individuals who have not been infected with this virus. The study involved 137 (men, n = 101; women, n = 36; mean age, 38.7; age range, 24–65 years) dentists and doctors from Kanagawa Dental University Hospital. Saliva and blood samples were analyzed by polymerase chain reaction (PCR) and immunochromatography for IgG and IgM, respectively. We then identified patients with saliva samples that were confirmed to be PCR-negative and IgM-negative for SARS-CoV-2. The cross-reactivity of IgA-positive saliva samples with SARS-CoV-2 was determined by enzyme-linked immunosorbent assay using a biotin-labeled spike recombinant protein (S1-mFc) covering the receptor-binding domain of SARS-CoV-2. The proportion of SARS-CoV-2 cross-reactive IgA-positive individuals was 46.7%, which correlated negatively with age (r = –0.218, p = 0.01). The proportion of IgA-positive individuals aged $\geq$50 years was significantly lower than that of patients aged $\leq$49 years (p = 0.008). SIgA was purified from the saliva of patients, which could partially suppress the binding of SARS-CoV-2 spike protein to the angiotensin converting enzyme-2 receptor. This study demonstrates the presence of SARS-CoV-2 cross-reactive SIgA in the saliva of individuals who had never been infected with the virus, suggesting that SIgA may help prevent SARS-CoV-2 infection.

**Funding:** The authors received from Kanagawa Dental University fund for this work. The funders had no role in study design, data collection and analysis, decision to publish, or preparation of the manuscript.

**Competing interests:** The authors have declared that no competing interests exist.

## Introduction

Secretory immunoglobulin A (SIgA) prevents infections through mucosal immunity—an aspect of the immune system. SIgA, comprising dimeric IgA, a J chain, and a secretory component, is secreted from glandular tissues such as the salivary glands and mammary glands onto mucosal surfaces, where it plays a central role in preventing the entry of antigens from the mucosa [1]. Severe acute respiratory syndrome coronavirus 2 (SARS-CoV-2) infects humans via the oral and nasal cavities, and the lungs [2]. The squamous cells of the tongue and periodontal tissues express angiotensin-converting enzyme-2 (ACE-2), a SARS-CoV-2 receptor, transmembrane protease serine 2 (TMPRSS2), and furin, which are proteases that promote infection [3], and saliva can harbor SARS-CoV-2 [4]. Saliva also contains several substances that suppress infection (such as lactoferrin, lysozyme, and SIgA, which is the most abundant) to potentially prevent the virus from entering the oral cavity [5].

Cross-reactive SIgA (CRsA) against SARS-CoV-2 was identified in breast milk before the COVID-19 pandemic [6, 7]. Furthermore, SARS-CoV-2-reactive CD4+ T cells were detected in approximately 40%–60% of unexposed individuals before the pandemic, suggesting that T cells have cross-reactivity to common cold coronaviruses and SARS-CoV-2 [8]. Later findings [9, 10] suggested that prior infection with coronavirus creates an immunological memory that is associated with IgG cross-reactivity.

Infection with SARS-CoV-2 causes COVID-19, which manifests as a unique spectrum of symptoms, ranging from asymptomatic to fatal acute respiratory failure [11]. The severity and prevalence of SARS-CoV-2 infection noticeably differs among age groups and countries [12]. Immune mechanisms might explain this wide disparity, but they are not yet fully understood. Immunoglobulin G (IgG) can eliminate SARS-CoV-2; thus, there is an urgent need for vaccine development against this virus [13]. However, mucosal immunity conferred by SIgA has not been investigated from the perspective of recovery from SARS-CoV-2 infection and its prevention. New findings in this area might facilitate a deeper understanding of COVID-19 characteristics.

Therefore, in this study, we aimed to develop an enzyme-linked immunosorbent assay (ELISA) for detecting SIgA that has cross-reactivity with SARS-CoV-2, and it could be used to reveal whether non-infected individuals harbor salivary SIgA that cross-reacts with the SARS-CoV-2 spike 1 subunit.

## Methods

### Participant selection

We tested saliva and blood samples using polymerase chain reaction (PCR) and immunochromatography, respectively. Individuals with saliva samples that were confirmed to be negative for COVID-19 by PCR and IgM testing were included in the study. The participants were 5 doctors and 132 dentists from Kanagawa Dental University Hospital. Individuals with IgA nephropathy, selective IgA deficiency, and autoimmune diseases, or who had cold-like symptoms within the past 2 weeks were excluded. The 137 participants, comprising 101 men and 36 women (mean age, 38.7; range, 24–65 years), provided fully informed consent. This study was approved by the Kanagawa Dental University Research Ethics Review Board (approval number: #690). This study was registered in the Japanese clinical trial UMIN-CTR (approval number: #R000046461) registry, which meets the ICMJE standards.

### Saliva collection for ELISA

We collected samples using Salivettes® (Sarstedt AG & Co., KG, Nümbrecht, Germany) in a fixed room of the hospital between 9 a.m. and 12 p.m. in August 2020 under infection control

protocols. The participants were instructed to refrain from eating, drinking, and brushing their teeth for at least 1 h before sample collection. The saliva samples were immediately centrifuged at $2,000 \times g$ for 15 min, and then stored at $-80°C$.

## Design of ELISA for CRsA against SARS-CoV-2 spike protein

We modified an ELISA system that could detect IgA cross-reactivity to influenza viruses using the human IgA ELISA quantitation set (#E88-102; Bethyl Laboratories, Montgomery, TX, USA) reported by Yamamoto et al. [14]. The saliva samples were diluted 500-fold in carbonate-bicarbonate buffer and incubated for 1 h at $25°C$. ELISA plate wells were washed five times with wash solution. The antigen was spike 1-mFc recombinant protein (#40591-V05H1; Sino Biological, Beijing, China) comprising the SARS-CoV-2 spike 1 subunit with the spike protein receptor-binding domain (RBD). The antigen was labeled with biotin using a kit as described by the manufacturer (#BK01; Dojindo Laboratories, Kumamoto, Japan). Biotin-labeled spike 1 was added to the ELISA plate with 1 µg/mL saliva sample per well and incubated for 1 h at $25°C$. The wells were washed five times with wash solution. Streptavidin-horseradish peroxidase conjugate (SA202; Millipore, USA; dilution, 1:1000) was then added to the wells and reacted for 1 h at $25°C$. TMB substrate solution was added to wells, allowed to react for 15 min at $25°C$, and the reaction was quenched with stop solution. Spike 1 protein-bound IgA was detected at 450 nm using a microplate absorbance reader (Bio-Rad Laboratories, Hercules, CA, USA). Background absorbance from the negative control containing phosphate-buffered saline was subtracted from the absorbance of all saliva samples.

## IgA purification and western blotting

We purified SIgA using an IgA purification kit (#20395; Thermo Fisher Scientific, Waltham, MA, USA) as described by the manufacturer. Purified saliva samples were added to the mixture of sample buffer (#NP0008; Thermo Fisher Scientific) and sample reducing agent (#NP0009; Thermo Fisher Scientific). The samples were heated for 5 min at $96°C$ and run on a sodium dodecyl sulfate-polyacrylamide gel electrophoresis (SDS-PAGE) gel using a standard protocol. The gel was stained with Coomassie brilliant blue (#178–00551; FUJIFILM Wako Chemicals, Osaka, Japan). Antibodies specific for the heavy chain of IgA were used to determine whether the purified substance was IgA. We further confirmed that the detected IgA was the secretory type (SIgA) using an antibody specific to the secretory component. Western blotting was employed using the following primary antibodies: anti-human IgA rabbit monoclonal antibody (ab184863; Abcam Plc, Cambridge, UK; 1:500 dilution) and anti-human IgA SC mouse monoclonal antibody (ab3924; Abcam, 1:500 dilution). Horseradish peroxidase-conjugated anti-rabbit polyclonal antibody (#P0448; Dako, Glostrup, Denmark; 1:1000 dilution) or anti-mouse monoclonal antibody (#P0447; Dako; 1:1000 dilution) was used as the secondary antibody.

## Ability of SIgA antibody to inhibit ACE-2-spike protein binding

To investigate whether ACE-2 binding to spike protein is inhibited by purified SIgA, we selected and pooled the top 20 antibody-positive and the bottom 20 antibody-negative samples based on the quantitative ELISA results. The final concentrations of antibodies in the pooled positive and negative saliva samples were 93.6 and 63.3 µg/mL, respectively.

The ability of SIgA to inhibit the binding of ACE-2 to the SARS-CoV-2 spike protein was then assessed using a SARS-CoV-2 spike-ACE-2 binding assay kit (#COV-SACE2-1; RayBiotech, Peachtree Corners, GA, USA), according to the manufacturer's instructions. A SARS-CoV-2 spike-neutralizing rabbit IgG monoclonal antibody (#40592-R001; Sino Biological) was

prepared as a positive control, and the neutralizing antibody was added at concentrations of 0, 0.0125, 0.025, 0.05, 0.1, and 0.2 µg/mL.

The percent binding inhibition (BI%) was determined as follows: [1 –(OD of test regent well/OD of no inhibitor in the positive control well)] × 100, according to the manufacturer's instruction.

### Questionnaire

The participants completed a self-administered questionnaire before saliva collection to determine whether they had previously been inoculated with Bacillus Calmette–Guérin (BCG), hepatitis B, and influenza vaccines within the past year. In addition, we requested the participants to fill out the questionnaire with their age and gender.

### Statistical analysis

Negative values were set to 0 to determine relative ELISA values of CRsA, and the association with age was analyzed using Spearman rank correlations. Associations between the relative positive ($>0$) and negative (0) values of CRsA in the ELISA and binary data, age group, sex, BCG vaccination status, hepatitis B vaccination status, and influenza vaccination status were examined. Associations with vaccination history were examined using chi-square or Fisher exact tests. We summarized the variables of hepatitis B and influenza vaccines as neither, either, or both to examine associations between the number of vaccinations and IgA positivity or negativity. The variables of BCG, hepatitis B vaccine, and influenza vaccine were summarized as zero, one, two, or three to examine associations between the number of vaccinations and IgA positivity or negativity. The significance level was set to 5%. All data were statistically analyzed using SPSS version 26 (IBM Corp., Armonk, NY, USA).

## Results

### CRsA against the SARS-CoV-2 spike protein

The relative value of CRsA determined using the ELISA was set to 0 as the negative value. The CRsA was positive in 64 (46.7%) and was negative (absorbance of 0 or below) in 73 (53.3%) of the 137 samples.

Fig 1 shows the plots showing age and cross-reactive IgA. Age significantly negatively correlated with relative CRsA ($r = -0.218$, $p = 0.01$). The positive rate of CRsA was significantly lower in participants aged $\geq 50$ years than in those aged $\leq 49$ years ($p = 0.008$), as shown in Table 1. The association between vaccines and CRsA positivity or negativity was not significant.

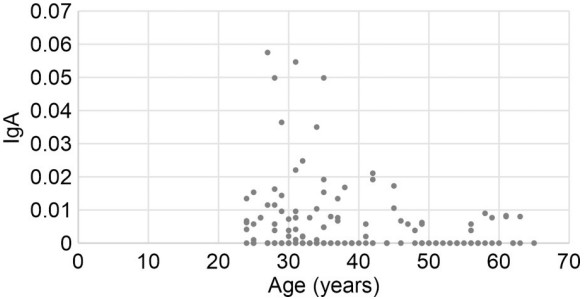

**Fig 1. Associations between participant age and positive or negative rate of CRsA.** Vertical axis: Relative amount of IgA determined as the absorbance of saliva samples from CRsA-positive participants. Negative samples are shown as 0. The absorbance was considerably high among many participants aged $\leq 49$ years.

**Table 1. Associations of CrSA positivity with age, sex, and vaccination status.**

| Variable | | Total | IgA-positive | | p* |
|---|---|---|---|---|---|
| | | n | n | % | |
| Age (years) | 20–29 | 32 | 19 | 59.4 | 0.081 |
| | 30–39 | 54 | 27 | 50.0 | |
| | 40–49 | 23 | 11 | 47.8 | |
| | 50–59 | 20 | 4 | 20.0 | |
| | ≥60 | 8 | 3 | 37.5 | |
| | 20–49 | 109 | 57 | 52.3 | 0.008 |
| | ≥50 | 28 | 7 | 25.0 | |
| Sex | Male | 101 | 48 | 47.5 | 0.452 |
| | Female | 36 | 16 | 44.4 | |
| Vaccination | | | | | |
| BCG | Yes | 102 | 49 | 48.0 | 0.781 |
| | No | 8 | 4 | 50.0 | |
| | Unknown | 27 | 11 | 40.7 | |
| Hepatitis B | Yes | 124 | 58 | 46.8 | 0.992 |
| | No | 11 | 5 | 45.5 | |
| | Unknown | 2 | 1 | 50.0 | |
| Influenza | Yes | 127 | 60 | 47.2 | 0.458 |
| | No | 10 | 4 | 40.0 | |
| Hepatitis B and Influenza | Neither | 1 | 1 | 100.0 | 0.114 |
| | Either | 19 | 7 | 36.8 | |
| | Both | 115 | 55 | 47.8 | |
| BCG, Hepatitis B, and Influenza | One | 3 | 1 | 33.3 | 0.809 |
| | Two | 21 | 11 | 52.4 | |
| | Three | 84 | 40 | 47.6 | |

*Chi-square or Fisher exact test.

## Characterization of purified saliva

Some IgA bands were detected in the purified saliva samples by SDS-PAGE (Fig 2, left panel). Signals of the secretory component (approximately 95 kDa), IgA heavy chain (approximately 60 kDa), IgA light chain (approximately 26 kDa), and J chain (approximately 13 kDa) were identified. Western blotting revealed a single band of approximately 60 kDa, confirming the presence of IgA heavy chain in the purified saliva (Fig 2, right panel). In addition, a single band of approximately 95 kDa was identified as the secretory component of SIgA (Fig 2, right panel). These confirmed that purified saliva contained SIgA components.

## CRsA inhibition tests

In the positive control, the range of absorbance values after the addition of spike-neutralizing antibody at different concentrations (0–0.2 μg/mL) was 2.155–0.493, with the concentration-dependent decrease confirming absorbance values. The neutralizing antibody against the spike protein showed an absorbance of 2.155 at an antibody concentration of 0 μg/mL. This absorbance indicates the absorbance when ACE-2 and the spike protein are strongly bound by ELISA. As the CRsA-negative saliva sample showed an absorbance of 2.402, which was higher than 2.155, it indicated that the binding between ACE-2 and the spike protein was not inhibited.

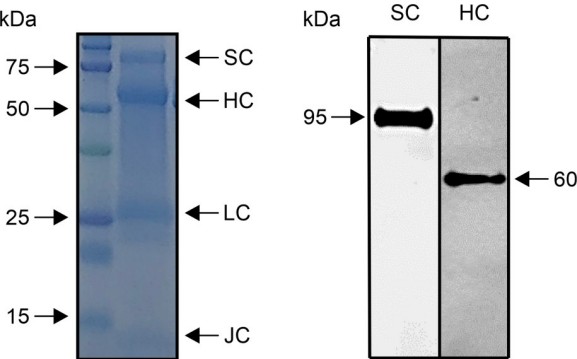

**Fig 2. Characterization of purified saliva.** The left panel shows protein bands of various molecular weights separated by SDS-PAGE. These bands were confirmed to correspond to the molecular weight of the secretory component (SC), IgA heavy chain (HC), IgA light chain, and J chain. The right panel shows the specific signals of SC and HC determined by western blotting.

The CRsA-positive saliva sample had an absorbance of 1.678, demonstrating the inhibition of binding between ACE-2 and the spike protein. The BI% was 21.7% for CRsA-positive saliva, but was −11% for CRsA-negative saliva. These results indicate that saliva SIgA1 antibody partially inhibited the spike protein.

## Discussion

In this study, we found that 46.7% of saliva samples from 137 participants who had not been infected with SARS-CoV-2 contained CRsA against SARS-CoV-2 spike 1. The spike 1 region that we examined included the RBD that binds to the SARS-CoV-2 receptor ACE-2. This region is important for preventing infections and could thus be a target for vaccine development [14]. A previous ELISA for detecting SARS-CoV-2 CRsA in breast milk showed positivity rates of 100% and 80% using total spike protein and RBD as the antigen, respectively [6]. However, the values of the saliva samples in the present study were lower than both values obtained in the previous study. This might be because more SIgA is produced in breast milk than in saliva [15]. The abundance of CRsA in breast milk is a proven post-vaccination response [16], which could further explain why the CRsA positivity rate was lower in saliva than breast milk.

The participants who tested negative for COVID-19 determined by PCR and antibody tests did not subsequently develop COVID-19 at the time of submission of this article. Thus, we believe that they had not been exposed to SARS-CoV-2. As such, we revealed SIgA antibodies against the spike 1 protein of SARS-CoV-2 in individuals with no history of SARS-CoV-2 infection. Only one other study found salivary IgA antibodies that cross-react with the spike protein of SARS-CoV-2 before the COVID-19 pandemic [17]. The nucleocapsid proteins of coronaviruses are highly homologous, but their spike proteins share little commonality [18]. Patients with a history of infection with HCoV-OC43 and other coronaviruses have antibodies against SARS-CoV-1 and antibodies that cross-react to the SARS-CoV-1 nucleocapsid protein, respectively [19]. Furthermore, while HCoV-NL63 uses ACE-2 as a receptor, its spike protein shares little homology with that of SARS-CoV-2 [20]. However, the cross-reactivity of receptor-binding motif 3 of NL63 and COV2-SPIKE$_{421-434}$ of SARS-CoV-2 has recently been reported [20]. As a past common coronavirus infection can induce antibodies that are cross-reactive to the spike and nucleocapsid proteins, it is reasonable that saliva would contain antibodies such as SIgA with cross-reactivity to SARS-CoV-2. However, the epitope responsible for salivary CRsA cross-reactivity remains to be clarified.

In the present study, we found that the CRsA level decreased with age. This might be explained by the fact that the IgA level decreases with age [21]. In addition, cross-reactive IgG antibodies were identified in 62%, 43.75%, and 5.72% of serum samples from individuals aged 1–16, 17–25, and ≥26 years, respectively, before the SARS-CoV-2 pandemic [9]. Because children are infected with common coronaviruses more frequently, this seems to indicate that more exposure leads to more individuals with cross-reactive antibodies. Moreover, cross-reactive T cells for SARS-CoV-2 are rare in elderly individuals [22]. These results can explain why CRsA is prevalent in younger individuals but uncommon in elderly individuals. This could offer a partial explanation to the mechanism underlying the fact that COVID-19 is less frequently severe and often asymptomatic in children [23] and adolescents [12].

IgA purified from saliva contains SIgA1, as determined by SDS-PAGE and western blotting. The purification kit used in this study is simple and versatile, but it has the limitation of not being able to measure IgA2 activity. In this study, we found that the binding between ACE-2 and spike protein was not inhibited in individuals without CRsA. In contrast, cross-reactive SIgA1 inhibited the binding of ACE-2 to the spike protein, demonstrating that SIgA1 functions as a neutralizing antibody against SARS-CoV-2. As blood IgA may prevent SARS-CoV-2 infection through a neutralization reaction [24], SIgA2 may have a similar function. Although this result demonstrates only partial suppression, this is the first report to indicate that SIgA may suppress SARS-CoV-2 in oral infection. Furthermore, if the neutralizing activity of SIgA containing SIgA1 and SIgA2 is investigated, the inhibition rate may increase.

Although saliva contains SARS-CoV-2, it also contains infection inhibitory factors [3]. Saliva lactoferrin is an infection suppressor that binds to SARS-CoV-2 [25]. Because SIgA has an antigen-processing function that is synergistic with lactoferrin, lysozyme, and peroxidase, salivary anti-bacterial or anti-viral factors might enhance the action of CRsA [26]. On one hand, early SARS-CoV-2-specific humoral responses are dominated by IgA antibodies, indicating that they play an important role in immunity after SARS-CoV-2 infection [27]. On the other hand, neutralizing IgA antibodies against SARS-CoV-2 persist in saliva for 49–73 days after the occurrence of symptoms [27]. Spike 1-CRsA in saliva is also associated with the severity of pneumonia in patients with COVID-19 [17]. Saliva can be collected easily and noninvasively, and SIgA is a practical test specimen because it is resistant to degradation and does not have strict transport conditions. Developing a method to easily and non-invasively measure SIgA in saliva might be important for the diagnosis or risk prediction of SARS-CoV-2 infection and responses to vaccines in the future.

A limitation of the present study was the low number of participants. Although vaccines stimulate the production of cross-reactive antibodies [7], we found no significant association between vaccines and cross-reactive antibodies. Future investigations should compare individuals who are not involved in medical care (without experience of vaccination).

## Conclusions

In this study, we identified the SARS-CoV-2 cross-reactive IgA spike protein in the saliva from individuals who did not have COVID-19. Elderly participants showed lower levels of SARS-CoV-2 spike protein-CRsA than younger participants. Salivary IgA might block the binding of ACE-2 to the spike protein. We revealed the importance of IgA as an inhibitor of SARS-CoV-2 infection in the oral cavity. Our findings are novel, as only a few studies have examined anti-SARS-CoV-2 antibodies in this compartment, despite the fact that the oral cavity is a recognized viral replication site [28].

## Supporting information

**S1 Fig. Original gel.** The slide shows the original gel image in Fig 2. Western blotting of (A) SDS-PAGE-separated components, (B) secretory protein, and (C) H chain; + indicates saliva purified with jacarin and–indicates unpurified saliva. Only saliva purified with jacarin is shown in Fig 2, which was cropped. No image processing was performed. M is a molecular weight marker.
(TIF)

**S2 Fig. Percentage of biding inhibition.** CRsA-positive saliva showed the neutralization activity for 21.72% (right bar). No apparent neutralization activity was observed in the CRsA-negative saliva (left bar).
(TIF)

## Acknowledgments

We are grateful to Ms. Makiko Yamada at the Research Support Center of Graduate School of Dentistry, Kanagawa Dental University for providing valuable technical assistance.

## Author Contributions

**Conceptualization:** Keiichi Tsukinoki.

**Data curation:** Tatsuo Yamamoto.

**Formal analysis:** Tatsuo Yamamoto.

**Investigation:** Juri Saruta.

**Methodology:** Keisuke Handa, Mariko Iwamiya.

**Project administration:** Satoshi Ino.

**Supervision:** Takashi Sakurai.

**Writing – original draft:** Keiichi Tsukinoki.

**Writing – review & editing:** Tatsuo Yamamoto, Keisuke Handa, Juri Saruta.

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
