## [Decision Letter · Decision Letter 0]

26 Apr 2021

PONE-D-21-10603

Detection of cross-reactive IgA in saliva against SARS-CoV-2 spike1 subunit

PLOS ONE

Dear Dr. Tsukinoki,

Thank you for submitting your manuscript to PLOS ONE. After careful consideration, we feel that it has merit but does not fully meet PLOS ONE’s publication criteria as it currently stands. Therefore, we invite you to submit a revised version of the manuscrPlease describe the characterization of the antibodies in detail.

Please describe the characterization of the antibodies in detail.

We look forward to receiving your revised manuscript.

Kind regards,

Etsuro Ito

Academic Editor

PLOS ONE

Journal Requirements:

PLOS requires an ORCID iD for the corresponding author in Editorial Manager on papers submitted after December 6th, 2016. Please ensure that you have an ORCID iD and that it is validated in Editorial Manager. To do this, go to ‘Update my Information’ (in the upper left-hand corner of the main menu), and click on the Fetch/Validate link next to the ORCID field. This will take you to the ORCID site and allow you to create a new iD or authenticate a pre-existing iD in Editorial Manager. Please see the following video for instructions on linking an ORCID iD to your Editorial Manager account: https://www.youtube.com/watch?v=_xcclfuvtxQ

4. Please amend the manuscript submission data (via Edit Submission) to include author Juri Saruta.

5. We note you have included a table to which you do not refer in the text of your manuscript. Please ensure that you refer to Table 1 in your text; if accepted, production will need this reference to link the reader to the Table.

6. Please ensure that you refer to Figure 1 in your text as, if accepted, production will need this reference to link the reader to the figure.

7. We note that you have included the phrase “data not shown” in your manuscript. Unfortunately, this does not meet our data sharing requirements. PLOS does not permit references to inaccessible data. We require that authors provide all relevant data within the paper, Supporting Information files, or in an acceptable, public repository. Please add a citation to support this phrase or upload the data that corresponds with these findings to a stable repository (such as Figshare or Dryad) and provide and URLs, DOIs, or accession numbers that may be used to access these data. Or, if the data are not a core part of the research being presented in your study, we ask that you remove the phrase that refers to these data.

Reviewers' comments:

Reviewer's Responses to Questions

**Comments to the Author**

1. Is the manuscript technically sound, and do the data support the conclusions?

Reviewer #1: Partly

Reviewer #2: Yes

2. Has the statistical analysis been performed appropriately and rigorously? 

Reviewer #1: Yes

Reviewer #2: Yes

3. Have the authors made all data underlying the findings in their manuscript fully available?

Reviewer #1: Yes

Reviewer #2: Yes

4. Is the manuscript presented in an intelligible fashion and written in standard English?

Reviewer #1: Yes

Reviewer #2: Yes

5. Review Comments to the Author

Reviewer #1: This manuscript simply assessed the cross-binding activity of salivary IgA antibodies to spike-1 protein of the SARS-CoV-2 virus. The findings are interesting; however, the authors should address several points in order to confirm the conclusion and to improve the quality of the manuscript.

Major Points:

1. It is essential to assess a more precise characterization of salivary IgA antibodies. For example, since the authors collected whole saliva samples from COVID-free participants, the distribution of monomeric versus polymeric forms of IgA antibody should be determined. Further, among the polymeric IgA antibodies, the distribution of dimeric and others should be analyzed.

2. Along the same line above, as the author described in the Discussion section, it is essential to assess the presence of other coronavirus-specific IgA antibody responses in saliva.

3. Since the IgA purification kit used in this study fails to collect IgA2 antibodies, the role of this subclass remains unknown. Since it has been shown that approximately up to 40 % of salivary IgA antibodies consist of the IgA2 subclass, it is essential to assess this subclass for the inhibition of spike-1 mediated binding to ACE-2.

4. The results of the inhibition of spike-1 mediated binding to ACE-2 by IgA antibodies should be presented in the figures or tables.

5. The authors described “purified IgA as a single band of ~60 kDa,” in the text (page 12, line 181). Since the molecular weight of secretory IgA (dimeric) is approximate ~390 kDa, even that of monomeric IgA is ~160 kDa, it is most likely that the authors employed unknown protein for the inhibition assay.

Minor Point:

1. Secretory IgA should be abbreviate as SIgA.

Reviewer #2: The manuscript by Lee et al. entitled “Detection of cross-reactive IgA in saliva against SARS-CoV-2 spike1 subunit” demonstrated the possibility that sIgA could bind to spike protein and prevent SARS-CoV-2 infection. It is of interest that participants of age 50 and over age 50 significantly showed a lower positive rate of CrsA than those of age 49 and under age 49. Moreover, the CrsA-positive sample can partially inhibit binding between ACE-2 and spike protein. The manuscript was clearly presented and well-written. The present information should be valuable for the clinician.

6. PLOS authors have the option to publish the peer review history of their article (what does this mean?). If published, this will include your full peer review and any attached files.

Reviewer #1: No

Reviewer #2: No

---

## [Author Response · Author response to Decision Letter 0]

1 Jul 2021

Response to the Editor:

IgA in human saliva exists as a secretory antibody and a serotype. Secretory IgA (SIgA) is secreted over the mucosal surface and protects the host against infection. Protease-resistant secretory component (SC) is required for IgA to function at the mucosal surface. Therefore, SIgA covers most of the mucosal surface, and IgA in saliva basically signifies SIgA. On the contrary, IgA without SC is the main component of serum. The ELISA used in this study was not specific to SIgA, but was specific to IgA and recognized SIgA as well. Although saliva also contains IgG, our ELISA specifically detected IgA in saliva and was unaffected by the presence of other antibodies. Nonetheless, SDS-PAGE and western blotting were performed to determine the characteristics of the detected antibody, and additional experiments revealed that IgA purified from saliva contained SIgA.　

Figure 1 was removed from the revised version since this was repeated data in the text. In the supporting information, original gels data was added.

1: This paper meets the style requirements of PLOS ONE.

2: I have obtained an ORCID.

3: The study participant's consent has been documented in the consent form; hence, I have clarified this aspect in the text and ethics statement field. Furthermore, no minors were included in the study.

4: I have added Saruta Juri to the list of authors.

5: "Table 1" and "Fig. 1" have been added to the text.

6: All data have been provided. Moreover, the original gel data have been provided as supporting data.

Response to the Reviewers:

Major Points

Response to Comment 1

Human saliva contains both SIgA secreted by the salivary glands and serum IgA. However, most of the IgA in saliva is considered to be SIgA. The concerned sample was that of a healthy individual who was negative for COVID-19, and I believe that the conventional concept of IgA applies here. Therefore, the distribution of SIgA and serum IgA in saliva was clear.

However, IgA in saliva purified using jacarin contained SIgA fragments, as CBB staining confirmed SC, light chain, heavy chain, and J chain signals. Furthermore, since the presence of SC is characteristic to the presence of a secretory antibody, western blotting of SC expression was performed to validate the signals noted through CBB staining. These results indicated that saliva purification using jacarin led to the detection of SIgA. The above-mentioned method (Line 122-136) and results (Line 191-204) have been added in the revised manuscript.

In addition, SIgA in saliva mainly exists as dimers. It has been reported that pentamer IgA is formed after intranasal vaccination; however, since these persons were not vaccinated, the distribution of antibody multimers was not evaluated.

Response to Comment 2:

As the reviewers point out, it makes sense to look at reactions towards other coronaviruses. On other hand, since it has been pointed out that the appearance of a cross-reactive antibody is related to the previous experience of common coronavirus disease, it is not surprising that cross-reactivity against SARS-CoV-2 occurs in saliva. 

It has been previously reported (Sci Immunol, ref 28) that the relationship between SARS-CoV-2 and the oral cavity (host) remains unclear. Thus, our study aimed to show whether cross-reactive antibodies against SARS-CoV-2 were present in host saliva. 

Since there is little knowledge about COVID-19, our findings, especially those pertaining to the age factor, are very important for determining the emergence of cross-reactive antibodies in COVID-19.

Response to Comment 3:

A jacarin purification kit was used to easily purify IgA from saliva. As the reviewers pointed out, IgA2 is not detected with this reagent kit. Here, the weaker neutralizing activity in the healthy subject may be due to the exclusion of IgA2 detection. However, it became clear, at least in this study, that IgA1 has neutralizing activity. In other words, our study demonstrated that IgA2 might also have neutralizing activity, and we believe that this is an important implication for IgA research. Therefore, the third paragraph of the discussion section has been significantly changed (Line 260-270). In the line 40, ‘partially’ was added, since data of IgA2 no included.

Response to Comment 4:

The binding inhibition rate (BI%) was determined according to the reagent kit manufacturer’s instructions using the formula: BI% = [1 – (OD of test reagent well/OD of positive control)] � 100. This information has been added to the method (Line 151-152), results (Line 216-218), and S2 (Line 412-414) in the revised manuscript.

Response to Comment 5:

The anti-IgA antibody used in the western blot analysis was prepared to target the constant region (250–350) of the heavy chain of human IgA as the antigen. Therefore, the signal generated in the western blot analysis represented the heavy chain of IgA, and the molecular weight was 55–60 kDa, which was not concerning. Furthermore, since SDS is used for sample processing, the S-S bonds were also separated.

In conclusion, since a singular fragment of the heavy chain was detected after SDS treatment, it can be considered that the western blot analysis recognized the IgA heavy chain.

Minor Points:

Response to Comment 1:

I have changed “sIgA” to “SIgA” at the relevant instances.

---

## [Decision Letter · Decision Letter 1]

13 Jul 2021

PONE-D-21-10603R1

Detection of cross-reactive Immunoglobin A against the Severe Acute Respiratory Syndrome-Coronavirus-2 spike 1 subunit in saliva

PLOS ONE

Dear Dr. Tsukinoki,

Thank you for submitting your manuscript to PLOS ONE. After careful consideration, we feel that it has merit but does not fully meet PLOS ONE’s publication criteria as it currently stands. Therefore, we invite you to submit a revised version of the manuscript that addresses the points raised during the review process.

Please provide your reply to the reviewer. I will decide my decision after carefully reading your reply.

We look forward to receiving your revised manuscript.

Kind regards,

Etsuro Ito

Academic Editor

PLOS ONE

Journal Requirements:

Reviewers' comments:

Reviewer's Responses to Questions

**Comments to the Author**

1. If the authors have adequately addressed your comments raised in a previous round of review and you feel that this manuscript is now acceptable for publication, you may indicate that here to bypass the “Comments to the Author” section, enter your conflict of interest statement in the “Confidential to Editor” section, and submit your "Accept" recommendation.

Reviewer #1: (No Response)

2. Is the manuscript technically sound, and do the data support the conclusions?

Reviewer #1: Partly

3. Has the statistical analysis been performed appropriately and rigorously? 

Reviewer #1: No

4. Have the authors made all data underlying the findings in their manuscript fully available?

Reviewer #1: Yes

5. Is the manuscript presented in an intelligible fashion and written in standard English?

Reviewer #1: No

6. Review Comments to the Author

Reviewer #1: Since the natural (antigen-non-specific) SIgA antibodies cross-reacted with Spike 1 subunit, it is significantly important to delineate the molecular characteristics of SIgA antibodies based on their polymeric styles. However, the authors failed to provide these essential results. The authors stated that salivary SIgA is mainly dimeric form without any direct evidence.

It is essential to assess the presence of common coronavirus-specific IgA antibody responses in saliva in order to interpret the outcomes of the current findings.

The authors failed to provide the role of IgA2 subclass antibodies for the inhibition of spike-1 mediated binding to ACE-2.

Binding inhibition assay was poorly performed. Since 20 of each positive and negative sample were assessed, individual results should be plotted with the mean values. Further, statistical analysis should be performed. The results should be moved into the main body.

7. PLOS authors have the option to publish the peer review history of their article (what does this mean?). If published, this will include your full peer review and any attached files.

Reviewer #1: No

---

## [Author Response · Author response to Decision Letter 1]

18 Sep 2021

Saliva is a valid biological fluid for testing the presence of SARS-CoV-2 mRNA. However, even though the oral cavity is the site of infection and viral replication of SARS-CoV-2, little is known about the local antigen-antibody response. Japan is presently in a state of emergency because of the delta strain.　Isho et al. (Science Immunol, 2020;5(52): eabe551.) investigated the persistence of salivary antibody responses to SARS-CoV-2 spike antigens in COVID-19 patients but there have been no reports of studies in healthy (non-infected) individuals. In particular, as vaccination progresses, findings in uninfected individuals will not be available due to the effects of the vaccine.

Our study examined the presence of crossed IgA antibodies against SARS-CoV-2 in uninfected individuals. To date, there have been few reports on this topic. Of particular interest is the association between age and crossed IgA antibodies: there was a significant difference in the presence of crossed IgA antibodies between individuals over 50 years old and those under 49 years old, with more crossed IgA antibodies in the younger individuals. This finding is consistent with the distribution of COVID-19 patients, suggesting that the measurement of crossed IgA antibodies may help predict susceptibility to infectious diseases in the future. Furthermore, the development of drugs that increase crossed IgA antibody levels could lead to the development of preventive and therapeutic drugs. This paper contains findings that are potentially important for the control of SARS-CoV-2; therefore, we request that it be accepted as soon as possible.

We have responded to the reviewers' questions as best as possible and have considered additional testing. However, owing to the spread of the delta strain, there are obstacles to various research activities, and additional testing is not possible. Even with the current research results, the paper is extremely important, and we ask for your favorable consideration.

Response to reviewers

We are very grateful to the reviewers for their careful peer review.

1. Salivary IgA is actively taken up by pIgRs in glandular atrial cells after dimeric IgA is produced by plasmacytoid cells, binds secretory components, and is discharged into conduits as sIgA. In the blood, IgA is predominantly monomeric, but on the mucosal surface, sIgA is mostly present (90% in parotid saliva). In addition, the secretory components (SC) results of western blotting showed the presence of sIgA, corresponding to the molecular characteristic mentioned by the reviewer.

2. A large amount of saliva is required to search for four or more common coronaviruses. However, due to the declaration of a state of emergency, it is very difficult to collect saliva. Past coronavirus infections may explain the presence of crossed IgA antibodies and this point has been well proven in previous studies as shown in the discussion.

3. As pointed out by the reviewer, the role of IgA2 has not been shown in the study, so this fact has now been included in the discussion as a limitation. 

4. 　In the binding inhibition assay, the test was performed using pooled saliva samples. The mean of the negative results is zero; we believe this does not allow for statistical analysis for comparison with positive results.

---

## [Editor Report · Decision Letter 2]

7 Oct 2021

Detection of cross-reactive Immunoglobin A against the Severe Acute Respiratory Syndrome-Coronavirus-2 spike 1 subunit in saliva

PONE-D-21-10603R2

Dear Dr. Tsukinoki,

We’re pleased to inform you that your manuscript has been judged scientifically suitable for publication and will be formally accepted for publication once it meets all outstanding technical requirements.

Kind regards,

Etsuro Ito

Academic Editor

PLOS ONE

---

## [Editor Report · Acceptance letter]

9 Nov 2021

PONE-D-21-10603R2 

Detection of cross-reactive Immunoglobulin A against the Severe Acute Respiratory Syndrome-Coronavirus-2 spike 1 subunit in saliva

Dear Dr. Tsukinoki:

I'm pleased to inform you that your manuscript has been deemed suitable for publication in PLOS ONE. Congratulations! Your manuscript is now with our production department. 

Kind regards, 

on behalf of

Prof. Etsuro Ito 

Academic Editor

PLOS ONE